# The Impact of Human Capital on Green Technology Innovation—Moderating Role of Environmental Regulations

**DOI:** 10.3390/ijerph20064803

**Published:** 2023-03-09

**Authors:** Jie Zhang, Shilong Li

**Affiliations:** 1School of Management Science and Real Estate, Chongqing University, Chongqing 400044, China; 2Research Center for Construction Economics and Management, Chongqing University, Chongqing 400044, China

**Keywords:** educational human capital, healthy human capital, environmental regulations, green technology innovation, moderating role

## Abstract

Green technology innovation can bring about dual benefits, i.e., technological progress and energy conservation, as well as emission reduction, which are regarded as effective means to achieve economic development and environmental protection. The influencing factors of green technology innovation have been studied from multiple angles. In order to promote the level of green technology innovation in China from a new perspective, this paper selected human capital as the independent variable, and empirically investigated the direct impact of educational and healthy human capital on green technology innovation, based on the panel data of 30 Chinese provinces (excluding Hong Kong, Macao, Taiwan and Tibet) from 2006 to 2016. Meanwhile, considering the current environmental policy system in China, this paper took environmental regulations as moderating variables, and analyzed the moderating role of three environmental regulations, namely, command-and-control environmental regulations, market-incentivized environmental regulations, and public voluntary environmental regulations, in the impact of human capital on green technology innovation. It was found that (1) educational human capital, with a three-period lag, and healthy human capital significantly promotes green technology innovation; (2) command-and-control environmental regulations, with a one-period lag, and market-incentivized environmental regulations promote green technology innovation, while public voluntary environmental regulations have an insignificant impact on green technology innovation; (3) the moderating effect of command-and-control and market-incentivized environmental regulations in the impact of human capital on green technology innovation is not significant. For public voluntary environmental regulations, the moderating effect between educational human capital and green technology innovation is significantly negative, while the moderating effect of healthy human capital on green technology innovation is not significant.

## 1. Introduction

China has achieved tremendous economic growth due to industrial development, as a result of reforms and the opening-up of the country. However, this rapid economic development has also brought about many negative impacts to China [1], such as environmental pollution, ecosystem degradation and resource depletion [2]. In this context, China has been actively looking for means to achieve, in parallel, economic development and environmental protection [3]. The central government of China has proposed that the basic state policy of resource conservation and environmental protection is unweaving, which demonstrates the country’s determination to achieve a win–win situation between “economy and environment”. Innovation is the fundamental driving force for economic and social development, and green technology innovation is the key to achieving low-carbon economic development and improving natural resource efficiency [4], which can yield the dual benefits of technological progress and energy conservation, as well as emission reduction. However, China’s green technology innovation is still at a relatively low level [5], and how to improve the level of green technology innovation has become one of the key issues in China.

Green technology innovation has long been studied from different perspectives. The concept was first proposed by Braun and Wield, defined as technology innovation that can achieve the goals of resource conservation, pollution emission reduction, and ecological protection [6]. Klassem et al. stated that green technology innovation places equal emphasis on the ecological environment and social development. Companies can achieve economic growth and environmental protection at the same time through green technology innovation [7]. In addition, there is a large body of literature that examines the influencing factors of green technology innovation. Reviewing this literature, it can be seen that factors influencing green technology innovation can be chiefly divided into two categories: external factors, such as environmental regulations [8,9] and media attention [10,11], and internal factors, such as business models used [12] and the corporate image [13]. Among them, the impact of environmental regulations on green technology innovation has been the focus of many studies.

The “Porter hypothesis” proposes that appropriate environmental regulations can promote green technology innovation [14]. In order to protect the environment and reduce pollution, the government will develop appropriate policies and regulations or resort to public awareness to restrain enterprises or individuals. In the face of these constraints, enterprises or individuals will weigh the gains and losses and consider green technology innovation [15]. There are numerous studies on the impacts of environmental regulations on green technology innovation, but the findings are inconsistent, and can be roughly summarized into two views; one is that environmental regulations can promote green technology innovation through the “innovation compensation” effect [16], while the other is that environmental regulations may inhibit green technology innovation due to the “crowding-out” effect [17].

In addition, talents are the main body of green technology innovation. The level of human capital in a region has an important impact on green technology innovation. Regional human capital is a fundamental driver of economic growth, which helps to reduce energy consumption by adopting new technologies and increasing productivity, bringing new economic value, and providing the main support and driving force for green technology innovation [18]. In recent years, the level of human capital in China has gradually improved, but it has not played its due role [19]. According to the education statistics in 2021, the gross enrollment rate of higher education in China has increased from 30% in 2012 to 57.8% in 2021; and the ratio of national financial expenditure on education to expenditures on GDP has reached 4.22% in 2020, which is the ninth consecutive year since 2012 to achieve a ratio of “no less than 4%”. The quality structure of the workforce has undergone significant changes, and the entire national workforce quality has been steadily improved. However, this high level of human resources has not led to a high level of innovation, and there is still a significant gap in innovation when compared with the United States, Japan and EU countries [20]. One possible reason might be that “factors related to social capital rather than human capital drive the acceptance of behaviors, products, and technologies that are individually costly but collectively beneficial” [21]. Studies have pointed out that an important reason for the under-utilization of human capital is the neglect of the heterogeneity of human capital investment structures. In existing research, human capital usually refers to educational human capital. However, in fact, health is also an important part of human capital [22]. As a result, the impact of human capital on green technology innovation might be underestimated, while the role of education on green technology innovation might be overestimated, with some attributing the impact of healthy human capital on green technology innovation to education instead [23]. In view of this, this paper divides human capital into two dimensions, educational and healthy, and analyzes the heterogeneous impact of the two types of human capital on green technology innovation.

Based on the above analysis, this paper empirically investigates the impact of human capital on green technology innovation under the perspective of environmental regulations. The research contributions of this paper are mainly as follows: (1) the role of human capital on green technology innovation has scarcely been examined, and the heterogeneity of human capital has rarely been taken into consideration in previous studies. To fill the gap, this paper classifies human capital into two categories, educational human capital and healthy human capital, and empirically investigates the mechanism of human capital’s influence on green technology innovation. (2) Existing studies have analyzed the “human capital–green development awareness” relationship [24] and the “environmental regulations–green technology innovation” relationship, but no studies have included human capital, heterogeneous environmental regulations and green technology innovation in the same research system. Therefore, a comprehensive research framework is established in this paper, classifying environmental regulations into three types: command-and-control, market-incentivized, and public voluntary environmental regulations. Then, the moderating role of different types of environmental regulations is explored in terms of the influence of human capital on green technology innovation, so as to provide policy recommendations for promoting green technology innovation and green development.

## 2. Literature Review and Research Hypothesis

Schultz’s human capital theory suggests that human capital promotes technological progress, which in turn generates more economic output and a higher efficiency. Romer’s endogenous growth theory emphasizes that the priority of human capital development is a key input for generating new ideas [25]. In the era of Industry 4.0, human capital, as a form of intellectual capital, is an important driving force for organizational innovation [26]. Among the components of human capital, education and health are the two most important factors. However, when studying the relationship between human capital and innovation, most scholars simply equate human capital with education [27,28]. Classical economic theories have pointed out that environmental pollution has significant negative externalities, and in order to uphold the principle of “who pollutes, who controls”, as well as to tackle environmental problems at the root, it needs to be ensured by government policies and regulations. Although the impact of human capital on green technology innovation and the impact of environmental regulations on green technology innovation have been investigated in previous studies, the three factors are rarely analyzed within one research framework. Therefore, this paper constructs a theoretically integrated framework of the impact of educational and healthy human capital on green technology innovation, and analyzes the moderating role of heterogeneous environmental regulations.

### 2.1. The Impact of Human Capital on Green Technology Innovation

Human capital consists of health, education, intelligence, training, skills, and other employment characteristics, as well as values or talents, such as punctuality and loyalty [29], which have economic value and are reflections of the quality of workers. Among them, health and education are the most important factors [30].

Knowledge, technology, ability and the experience of green technology innovation embedded in educational human capital play an important role in green technology innovation. Education has a significant positive impact on the green economy [31], which is realized mainly in three ways. First, a high level of educational human capital can increase public awareness towards environmental protection and energy use, thereby cultivating high-quality green technology innovation talents for enterprises. Enterprises with a well-educated labor force are more inclined to implement environmental standards, enhance environmental protection [32], and transform R&D technology to be more energy-efficient [33], thereby promoting green technology innovation. Second, a higher level of educational human capital often means a higher level of income and environmental protection concepts, corresponding to a higher quality of local consumers. Guiding local residents towards low carbon consumption rates, low-pollution industries and environmentally friendly living will have a positive impact on the green development of society, which is conducive to the green choices made by the “consumption side”, as well as to improving the efficiency of the green economy [34]. In addition, people with high educational human capital tend to be aware of environmental damage, and they will spontaneously establish environmental organizations to pressure governments and enterprises to find ways to reduce resource waste [35]. Third, educational human capital also has spillover properties. Cities with above-average levels of human capital may have more knowledge spillover, making it easier to elicit new knowledge [36]. Additionally, it can also enable the rapid diffusion of advanced foreign technologies, thereby promoting local enterprises to innovate green technology under high environmental standards [37].

Unlike educational human capital, which cultivates green awareness among citizens and nurtures green technology innovators for enterprises, healthy human capital plays a fundamental role in green technology innovation. Healthy human capital influences green technology innovation in three ways. First, by providing more productive energy, a high level of healthy human capital enables workers to be physically and mentally strong enough to increase their labor efficiency, which lays a solid foundation for green technology innovation. Second, healthy human capital can improve workers’ learning efficiency and cognitive abilities, which indirectly improves the returns to education, and increases the accumulation of educational human capital, ultimately positively influencing green technology innovation in an indirect way [38,39]. Third, good health is an important goal that the public has always been pursuing. By investing more in health care, enterprises can maintain a high level of healthy human capital, thereby providing employees with a healthy work environment. On this basis, employee training can be better developed to improve the comprehensive quality of employees, and further improve the green technology innovation ability of enterprises [40]. Based on the above review, this paper proposes Hypothesis 1 and Hypothesis 2:

**H1.** *Educational human capital positively influences green technology innovation*.

**H2.** *Healthy human capital positively influences green technology innovation*.

### 2.2. The Impact of Environmental Regulations on Green Technology Innovation

Environmental regulation, one of the topical issues in environmental economics, is widely present in current academic research, and the following views have been developed:

Empirical studies have verified that environmental regulations have a catalytic effect on green technology innovation. As a policy tool, governments often use environmental regulations to force enterprises to change their original production processes, products and technologies, as well as to reduce pollutant emissions [41]. Environmental regulations lead to higher pollution control costs, and enterprises will increase their R&D expenditure to maintain production costs and profitability, thereby promoting the green transformation of production processes [42,43]. If environmental regulations are stringent enough, it is a must for enterprises to develop new products that consume less energy and cause less pollution in order to remain competitive. When compared to other industries, heavily polluting industries face stricter environmental regulations, so they have a greater number incentives to use or develop new technologies [44].

However, other scholars, contrary to the above view, argue that environmental regulations inhibit green technology innovation through a “compliance cost” effect. Under stringent environmental regulations and high environmental standards, enterprises often spend limited funds on preventing and reducing pollution emissions, thereby crowding out capital for green technology innovation activities and inhibiting green technology innovation [45,46]. In addition, under the new Environmental Protection Law, enterprises are strongly biased against investment. In the face of higher pollution discharge standards, enterprises will reduce environmental pollution in the production process through end-of-line treatment and relocation of pollution departments, which will increase the fixed investment of enterprises and crowd out R&D investment. As a result, the development of technologies in the production process will be negatively affected, and thus inhibiting green technology innovation [47].

In addition to the linear relationship, it is found that there more complicated relationships also exist. Some scholars believe that there is an inverted U-shaped relationship between environmental regulations and green technology innovation. When the regulatory intensity is low, green technology innovation will increase with the increase in the intensity of environmental regulations; however, when the intensity of environmental regulations reaches a certain point, the level of green technology innovation begins to decline [48,49]. From a dynamic perspective, Peuckert empirically demonstrates that environmental regulations have short-term inhibitory, but long-term positive, effects on productivity growth [50]. Some studies classify environmental regulations into three types: command-and-control environmental regulations, market-incentivized environmental regulations and public voluntary environmental regulations, with different types of environmental regulations presenting different impact relationships [51]. Command-and-control environmental regulations not only strengthen the economic penalties for violations, but also increase the administrative detention penalties, which greatly increases the cost of environmental violations and forces enterprises to engage in green technology innovation [52]. Market-incentivized environmental regulations are formulated based on the principle of “who pollutes, who pays”. When enterprises face high environmental costs, they tend to upgrade their technologies to reduce pollution [53]. Public voluntary environmental regulations rely on public awareness to discipline enterprises; however, the positive effect is limited and weak, because no adequate pressure is exerted [54].

This paper argues that the incentive effects of green technology innovation differ significantly between different types of environmental regulations, in terms of their mechanism of action, prerequisites and pathways. Accordingly, this paper proposes Hypothesis 3:

**H3.** *Different types of environmental regulations play different roles in influencing green technology innovation*.

Command-and-control environmental regulations positively affect green technology innovation; market-incentivized environmental regulations promote green technology innovation; public voluntary environmental regulations have insignificant effects on green technology innovation.

### 2.3. The Moderating Role of Different Types of Environmental Regulations in the Impact of Human Capital on Green Technology Innovation

The theory of institutional regulation refers to regulating organizational behavior through economic and social regulation. Enterprises are inevitably affected by environmental regulatory policies when carrying out pollution emission and green technology innovation. From the above analysis, it can be seen that human capital will greatly affect the level of green technology innovation of enterprises, and the intensity of environmental regulations may affect the level of human capital investment to some extent. Therefore, this paper argues that environmental regulations have a moderating role in the impact of human capital on green technology innovation.

Under formal environmental regulations, human capital can reduce the relative amount of physical capital and the degree of capital–factor mismatch, enable enterprises with backward technology and substandard emissions to exit, and optimize the overall capital–factor allocation in the industry, thus promoting green total factor productivity improvement [55]. Due to stringent command-and-control environmental regulations, some enterprises have to invest a large amount of money in pollution prevention and control in the short term, in order to avoid administrative penalties. This will lead to a decrease in human capital investment and inhibit the role of human capital in promoting green technology innovation [56]. When market-incentivized environmental regulations increase, in order to reduce pollution costs and meet local environmental standards, enterprises will increase human capital investment by training their own employees and bringing in innovative talents [57], in order to enhance the influence of human capital on green technology innovation. In addition, the environmental pollution of enterprises will directly affect the normal life and health of residents in nearby communities [58]. Therefore, in order to quickly reduce the impact of pollution on the public, enterprises may invest a large amount of funds in pollution control strategies, thus crowding out a portion of human capital investment and weakening the influence of human capital on green technology innovation. Accordingly, this paper proposes Hypothesis 4:

**H4.** *Different types of environmental regulations play different moderating roles in the impact of human capital on green technology innovation*.

Command-and-control environmental regulations and public voluntary environmental regulations play a negative moderating role, while market-incentivized environmental regulations play a positive moderating role between human capital and green technology innovation.

The theoretical analysis framework of this paper is illustrated in Figure 1:

## 3. Research Design

### 3.1. Variable Selection and Data Sources

This paper selected the period comprising the deepening, stabilizing and maturing of China’s environmental policy, from 2006 to 2016, as the study period. During this period, China’s environmental regulations were more widely applied and had a more far-reaching impact. Selected in terms of data availability, 30 provinces in China (excluding Hong Kong, Macao, Taiwan and Tibet) were selected for the study, and the missing data are linearly interpolated. With green technology innovation as the dependent variable, human capital as the independent variable, and environmental regulations as the moderating variable, the impact of human capital on regional green technology innovation under the perspective of different types of environmental regulations was examined.

The data of green technology innovation in this paper were obtained from the “ccer” database, and other data were obtained from the China Environment Yearbook and China Statistical Yearbook. The specific measurements of each variable are as follows:(1)Dependent variable

Green technology innovation (GTI): There are three main ways to measure green technology innovation. Firstly, it is measured by the ratio of enterprises’ R&D investment to their total energy consumption, and the larger the ratio, the stronger the green technology innovation [59]. Secondly, stochastic frontier analysis (SFA) is used to measure green technology innovation efficiency [60]. The third strategy is the use of green patent statistics [61,62,63]. Although patents are hard to be fully and effectively applied to the social production process to create benefits, it is undeniable that they are a key output of the innovation process and an important manifestation of innovation outcomes. In the available statistics, green patents consist of grants and applications [64]. In this paper, the total number of green patent grants was used to measure the level of green technology innovation.

(2)Independent variable

Educational human capital (EHC): Currently, two main strategies are used to measure educational human capital, i.e., the average years of schooling, and the share of local financial expenditure on education in the regional GDP. In this paper, the latter was chosen to measure educational human capital.

Healthy human capital (HHC): Healthy human capital, as an important component of human capital, is closely related to the level of healthcare services and healthcare expenditure enjoyed by economic individuals [65]. Government health expenditure helps to improve intergenerational income transmission and promote healthy human capital accumulation. To exclude the effect of individual income [66], this paper chose the regional GDP’s share of local financial expenditure on healthcare to measure the level of healthy human capital.

(3)Moderating variable

The current methods used to analyze environmental regulations mainly include the following types: one uses a single indicator to measure environmental regulations, such as the emission density of a certain pollutant [67], the number of environmental administrative punishment cases [68], and the total investment in environmental pollution control multiplied by the corresponding weighting coefficient [69], etc. However, it is hard using a single indicator to comprehensively and accurately measure the effect of the environmental regulations. The second method is to specifically select the emission levels of industrial tertiary wastes as different original indicators, reflecting environmental pollution, and to calculate the intensity of comprehensive environmental regulations with an improved entropy method [70]. The third method is to divide environmental regulations into different types [71], and consider the different types of environmental regulations as having different impacts.

This paper, referring to the classification methods of environmental regulations by Yan Ying [72], divides environmental regulations into the following three types:

Command-and-control environmental regulations (ER1): total investment in pollution control/GDP.

Market-incentivized environmental regulations (ER2): this is measured by the amount of fees paid to the national treasury for waste discharge per capita, in each province.

Public voluntary environmental regulations (ER3): an emerging environmental regulation, which means that citizens voluntarily participate in environmental protection, and exert pressure and supervision on the pollution emissions of enterprises. The total number of environmental petitions in each region is used to measure public voluntary environmental regulations in this paper.

(4)Control variables

Four variables are selected as control variables, each of which has a certain impact on green technology innovation. The four control variables are measured as follows:

Regional economic development level (PGDP): economic development financially guarantees green technology innovation. Generally speaking, the higher the level of economic development, the more excellent talents and advanced technologies can be brought to enterprises. PGDP is measured by GDP per capita in this paper.

Industrial structure (STR): measured by the proportion of the added value of secondary industry to the total output value of a region. Although the secondary industry has enabled China’s rapid economic development, it has also brought about serious environmental pollution problems. Lowering the proportion of the secondary industry is more conducive to green technology innovation.

R&D intensity (RD): to a certain extent, this reflects the intensity of R&D investment in green technology innovation. This is measured by internal expenditure on R&D per capita.

Urbanization level (UR): the level of urbanization can reflect the concentration of labor and technical personnel in a region, which provides a guarantee of manpower for green technology innovation. The ratio of the urban population to the total population in each region is applied in this paper to measure the urbanization level.

Table 1 visually describes how variables are measured:

The relationship between the elements is shown in Figure 2:

### 3.2. Model Selection

At present, the following methods are used to analyze green technology innovation: (1) regression analysis using ordinary least squares [73,74,75]; (2) game models, in order to test the impact of environmental regulations on green technology innovation [76]; (3) DID models, in order to study the impact of environmental standards on green technology innovation [77].

Considering the number of variables and the form of the data, this paper first used a multiple linear regression model for analysis. Before conducting the regression analysis, a Hausman test was performed to determine whether fixed effects or random effects should be used in the model. The results indicate a *p*-value of 0.0000, which strongly rejects the original hypothesis that the disturbance term is not correlated with individual characteristics; so, the dual fixed-effects model was chosen for the following regression analysis.

Taking the actual situation into account, this paper develops the following three panel data models to investigate (1) the impact of human capital on green technology innovation; (2) the impact of environmental regulations on green technology innovation; (3) the moderating role of environmental regulations in the impact of human capital on green technology innovation.
(1)$$GTI_{it}=\alpha_{0}+\beta_{1}HC_{it}+\gamma Control_{it}+\mu_{i}+\nu_{t}+\varepsilon_{it}$$
(2)$$GTI_{it}=\alpha_{1}+\beta_{2}ER_{it}+\gamma Control_{it}+\mu_{i}+\nu_{t}+\varepsilon_{it}$$
(3)$$GTI_{it}=\alpha_{2}+\beta_{1}HC_{it}+\beta_{2}ER_{it}+\beta_{3}(HC_{it}\;\times \;ER_{it})+\gamma Control_{it}+\mu_{i}+\nu_{t}+\varepsilon_{it}$$

$\beta_{1},\beta_{2,}\;\mathrm{and}\;\beta_{3}$ represent the parameters to be estimated; $\alpha_{0},\;\alpha_{1},\mathrm{and}\;\alpha_{2}$ are constant terms; $Control_{it}$ is a series of control variables; γ is the parameters of control variables; $\mu_{i}$, and $\nu_{t}$ represent regional fixed effects and time fixed effects, respectively; $\varepsilon_{it}$ represents random disturbance terms. i and t indicate the province and year, respectively. Model (1) mainly examines the differentiated impact of human capital on green technology innovation; Model (2) examines the impact of different types of environmental regulations on green technology innovation; and Model (3) introduces the interaction between environmental regulations and human capital on the basis of Models (1) and (2), which was used in order to test the moderating effect of different types of environmental regulations on human capital and green technology innovation.

## 4. Empirical Analysis

### 4.1. Descriptive Statistics

Results of descriptive statistics for the main variables are shown in the Table 2.

### 4.2. Overall Estimation Results

The regression results are shown in Table 3:

From the regression results, it can be seen that the level of educational human capital and the level of healthy human capital are positively correlated with green technology innovation, which verifies Hypothesis 1 and hypothesis 2. Moreover, the relationship between educational human capital and green technology innovation in the current period is not significant, while educational human capital, with a three-period lag, significantly promotes green technology innovation. This may be due to the fact that there is a certain lag in terms of human participation in social activities organized to produce economic value after the investment in educational resources. Healthy human capital significantly affects regional green technology innovation. It is an important premise and foundation for other human capital investments, which can enhance the physical fitness of workers, improve their labor efficiency, and ultimately, increase the rate of returns on educational human capital investment. In addition, a high level of healthy human capital can provide employees with a healthy working environment, thereby improving the comprehensive quality of employees, improving their ability to use technology, and further promoting green technology innovation.

The results also indicate that different types of environmental regulations have different effects on green technology innovation, which verifies Hypothesis 3.

Command-and-control environmental regulations, with a one-period lag, significantly promote green technology innovation. The reason may be that since current command-and-control environmental regulation tools are strongly enforced, in order to meet the government’s environmental protection requirements in the short term, enterprises are likely to use R&D funds for pollution control, which is not conducive to carrying out green technology innovation. However, over time, the command-and-control environmental regulations will gradually improve, guiding enterprises to solve the pollution problem at the source, which will dramatically reduce the risk of the failure of green technology innovation in enterprises, and consequently, facilitate their green technology innovation [78].

Market-incentivized environmental regulations significantly promote green technology innovation. This may be ascribed to the fact that market regulations, such as levying emission fees, give enterprises an impetus to develop green technology innovation, which sufficiently stimulates the enthusiasm of enterprises for green technology innovation. Enterprises engage in pollution control not only for the purpose of meeting the government’s environmental protection requirements, but also for their own interests and development. Therefore, compared with end-of-line treatment, which is a palliative behavior, enterprises are more willing to obtain government incentive resources for green technology innovation, which is beneficial to the long-term stable development of enterprises [79].

The impact of public voluntary environmental regulations on green technology innovation is not significant. This may be explained by the fact that public participation, as a means of environmental regulations, has not emerged for a long time, and is not strong enough to constrain enterprises. Most enterprises usually resort to end-of-line governance when faced with public supervision. In general, public concern about the environment will influence environmental behavior only when the cost is low and no inconvenience is caused [80]. Moreover, the cost of adopting new technologies may be another reason. Green products, though being more economical in the long run, tend to be more expensive, and the public may prefer to buy non-green products [81]. This would discourage enterprises from engaging in green technology innovation. What is worse, in the absence of clear information about particular environmental behaviors, people tend to underestimate the involvement of others. Such bias weakens the public’s willingness to take environmental actions [82], and, therefore, public voluntary environmental regulations have no significant impact on the level of green technology innovation of enterprises.

### 4.3. Analysis of Moderating Effects

The regression results of the moderating effects are shown in Table 4:

The regression results show that both command-and-control environmental regulations and market-incentivized environmental regulations play a positive moderating role between the two types of human capital and green technology innovation; however, the results are not significant. Public voluntary environmental regulations have a significantly negative moderating effect between educational human capital and green technology innovation, and an insignificantly negative moderating effect between healthy human capital and green technology innovation. Hypothesis 4 is verified.

The possible reason for this is that when the intensity of command-and-control and market-incentivized environmental regulations increases, enterprises will increase their investment in the green technology innovation R&D process to a greater extent, and they will not crowd out or increase investment in the source of green technology innovation R&D. As a result, there is no significant moderating effect between human capital and green technology innovation.

Public voluntary environmental regulations play a negative moderating role between human capital and green technology innovation. This may be because when the public voluntary environmental regulations reach a certain intensity, enterprises are forced by public pressure to meet public environmental standards in a short period of time. However, green technology innovation is a long-term process. Therefore, in order to meet requirements while maintaining their current production statuses, enterprises have to choose end-of-line treatment, which requires them to transfer part of their funds to pollution control, thereby crowding out human capital investment and weakening its role in promoting green technology innovation.

## 5. Robustness Tests

In addition to the dual fixed-effects model, a negative binomial regression model was chosen as an alternative to elucidate the robustness of the basic regression results. The results are shown in Table 5, which indicates that human capital promotes green technology innovation, with ER1 and ER2 positively affecting green technology innovation, and ER3 having an insignificant effect.

The robustness test was also conducted by replacing the core dependent variables [83]. The number of green utility patents obtained (GUP) was used as a proxy for green technology innovation (GTI) for the robustness test. The number of green utility patents obtained not only reflects the utility level of regional green technology innovation, but also is a major means that is used to meet the public demand for green consumption. Results of the robustness test are shown in Table 6. It can be seen that the results are consistent with the above-mentioned research results, which again verifies the reliability of previous studies.

## 6. Conclusions

### 6.1. Research Conclusions

This study enriches the existing literature. First, this paper divides human capital into two categories, educational and healthy human capital, and explores the mechanisms of their effects on green technology innovation. Government support, such as that given through technology, R&D, and education, directly or indirectly encourages open innovation in green technology innovation [84]. Educational human capital promotes green technology innovation by cultivating highly qualified green technology innovation talents [32], developing environmental protection concepts [33], and accelerating the diffusion of advanced technologies [37]. Healthy human capital enhances the physical fitness of workers and improves their learning efficiency, thus increasing the educational returns to human capital, which indirectly promotes green technology innovation [38,39]. Second, this paper incorporates different types of environmental regulations, human capital, and green technology innovation into the same research framework to investigate the moderating role of environmental regulations between human capital and green technology innovation, and briefly outlines an overview of the moderating mechanism: the implementation of environmental regulations moderates the effect of human capital on green technology innovation, by increasing or decreasing the input of human capital.

Selecting educational human capital, healthy human capital and three environmental regulations, i.e., command-and-control environmental regulations, market-incentivized environmental regulations and public voluntary environmental regulations, as variable indicators, this paper establishes fixed effects and moderating effects models to empirically investigate the direct impact of human capital on green technology innovation, and the moderating role of different types of environmental regulations in the impact of human capital on green technology innovation. Data were obtained from the panel data of 30 Chinese provinces (excluding Hong Kong, Macao, Taiwan, and Tibet), from 2006 to 2016. It was found that: (1) educational human capital, with a three-period lag, and healthy human capital significantly promotes green technology innovation; (2) command-and-control environmental regulations, with a one-period lag, and market-incentivized environmental regulations promote green technology innovation, while the effect of public voluntary environmental regulations on green technology innovation is not significant; (3) the moderating effect of command-and-control and market-incentivized environmental regulations in the impact of human capital on green technology innovation is not significant. For public voluntary environmental regulations, the moderating effect between educational human capital and green technology innovation is significantly negative, while the moderating effect between healthy human capital and green technology innovation is not significant.

### 6.2. Policy Recommendations

Based on the above findings, this paper puts forward the following policy recommendations:

(1) Increase investment in educational human capital and healthy human capital to promote an increased level of regional green technology innovation at the source.

At this stage, the scale of China’s education has entered the forefront of the world, but the utility of human capital has not been effectively utilized. Therefore, the government should increase investment in education to vigorously develop higher education, foster citizens’ awareness of green environmental protection, and cultivate green technology innovation talents for enterprises. In addition, the government should also formulate corresponding policies to attract outstanding green technology innovation talents, in order to improve the level of regional green technology innovation. Enterprises should attach importance to the on-the-job education of employees and launch regular employee training for continuously improving the green technology innovation literacy of employees. Moreover, enterprises should also establish an incentive mechanism for green technology innovation and encourage employees to conduct green technology innovation.

Good physical conditions are the basic guarantee for talents to carry out green technology innovation. The government should strengthen the construction of public health services and increase investment in public health, so as to provide citizens with a healthy working and living environment. In addition, the government should also encourage citizens to engage in physical exercises to enhance their physical fitness, which is beneficial for ensuring that the educational human capital will be well utilized, as well as laying a good foundation for green technology innovation.

(2) Adopt reasonable environmental regulations and formulate appropriate environmental regulation policies.

From the above analysis, it can be seen that command-and-control and market-incentivized environmental regulations significantly promote green technology innovation. To improve the level of regional green technology innovation, the government should make full use of these two environmental regulation tools, grasp the strength of command-and-control environmental regulations, and pay more attention to the long-term stable adjustment effect of market-incentivized environmental regulations.

Public voluntary environmental regulations play a significantly negative moderating role between educational human capital and green technology innovation. Therefore, the government should make rational use of public voluntary environmental regulations, in order to seek a balance between them and human capital. Government subsidies can be granted when necessary to ensure the maximum promotion of green technology innovation.

## Figures and Tables

**Figure 1 ijerph-20-04803-f001:**
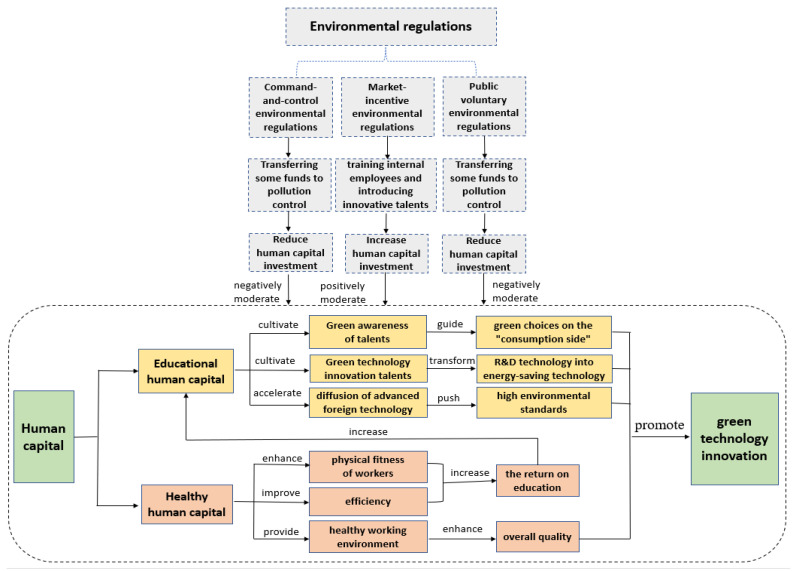
Theoretical analysis framework.

**Figure 2 ijerph-20-04803-f002:**
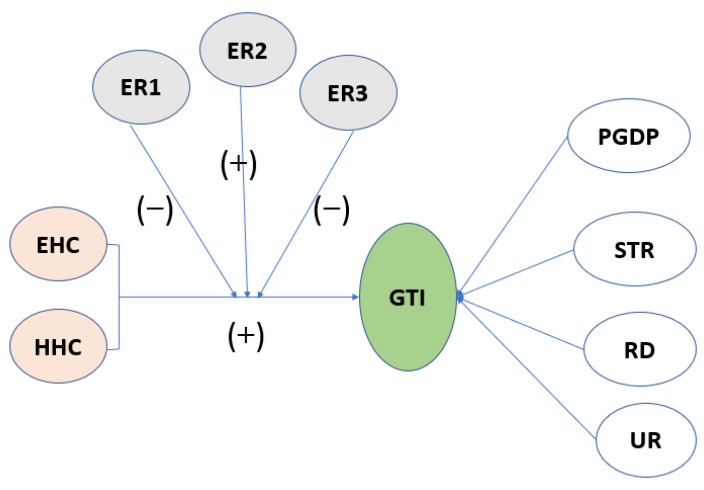
The relationship between the elements. (Note: “(+)” in the figure indicates positive promotion and “(−)” indicates negative inhibition).

**Table 1 ijerph-20-04803-t001:** Metrics table.

	Name	Symbols	Methods
Dependent variable	Green technology innovation	GTI	the total number of green patent grants
Independent variable	Educational human capital	EHC	local financial expenditure on education/GDP
	Healthy human capital	HHC	local financial expenditure on health care/GDP
Moderating variable	Command-and-control environmental regulations	ER1	total investment in pollution control/GDP
	Market-incentivized environmental regulations	ER2	fees paid to the national treasury for wastes discharge per capita
	Public voluntary environmental regulations	ER3	the total number of environmental petitions
Control variables	Regional economic development level	PGDP	GDP per capita
	Industrial structure	STR	the proportion of the added value of secondary industry/GDP
	R&D intensity	RD	internal expenditure on R&D per capita
	Urbanization level	UR	urban population/total population

**Table 2 ijerph-20-04803-t002:** Results of descriptive statistics.

	Name	Obs.	Mean	Std. Dev.	Min	Max
Dependent variable	GTI	330	6.596584	1.529992	1.609438	9.786392
Independent variable	EHC	330	3.56115	1.400396	1.323908	9.073244
	HHC	330	1.464715	0.7414536	0.3321076	4.113692
Moderating variable	ER1	330	1.378576	0.6856546	0.3	4.24
	ER2	330	14.14346	10.21269	0.6508068	81.5985
	ER3	330	8.342095	1.623557	0	11.65609
Control variables	PGDP	330	10.39064	0.5899677	8.71	11.68
	STR	330	47.18036	8.03955	19.26	61.5
	RD	330	720.5055	1034.22	25.17	6831.92
	UR	330	52.90115	13.82496	27.47	89.61

(Note: GTI—Green technology innovation; EHC—Educational human capital; HHC—Healthy human capital; ER1—Command-and-control environmental regulations; ER2—Market-incentivized environmental regulations; ER3—Public voluntary environmental regulations; PGDP—Regional economic development level; STR—Industrial structure; RD—R&D intensity; UR—Urbanization level).

**Table 3 ijerph-20-04803-t003:** Regression Results of the Impact of Human Capital and Environmental Regulations on Green Technology Innovation.

	Model (1)			Model (2)		
EHC	0.0230 (0.80)					
L3.EHC		0.0569 * (1.90)				
HHC			0.249 *** (3.72)			
L.ER1				0.0442 * (1.82)		
ER2					0.00414 * (1.96)	
ER3						0.00586 (0.54)
PGDP	−0.176 (−1.00)	0.047 (0.20)	−0.243 (−1.42)	−0.115 (−0.63)	−0.287 (−1.59)	−0.206 (−1.17)
STR	0.0167 *** (3.73)	0.0212 *** (3.47)	0.0170 *** (3.90)	0.0182 *** (3.87)	0.0158 *** (3.55)	0.0167 *** (3.73)
RD	0.000178 *** (4.54)	0.0000766 (1.52)	0.000257 ** (5.86)	0.000152 *** (3.81)	0.000156 *** (4.08)	0.000218 *** (4.51)
UR	0.0659 *** (7.96)	0.0448 *** (4.09)	0.0657 *** (8.13)	0.0597 *** (6.85)	0.0687 *** (8.24)	0.0658 ** (7.94)
_cons	2.911 * (1.96)	2.155 (1.96)	3.405 * (2.38)	2.865 * (1.84)	3.897 * (2.04)	3.206 ** (2.18)
N	330	240	330	300	330	330
R2-within	0.9617	0.9417	0.9634	0.9571	0.9621	0.9616
R2-between	0.2839	0.2573	0.2233	0.2927	0.2830	0.3023
R2-overall	0.4788	0.3690	0.4317	0.4563	0.4784	0.4935
F-value	476.59	266.62	499.45	407.53	482.13	475.98
*p*-value	0.0000	0.0000	0.0000	0.0000	0.0000	0.0000

(Notes: t-statistics in parentheses. *** *p* < 0.01. ** *p* < 0.05. * *p* < 0.1. GTI—Green technology innovation; EHC—Educational human capital; HHC—Healthy human capital; ER1—Command-and-control environmental regulations; ER2—Market-incentivized environmental regulations; ER3—Public voluntary environmental regulations; PGDP—Regional economic development level; STR—Industrial structure; RD—R&D intensity; UR—Urbanization level).

**Table 4 ijerph-20-04803-t004:** Regression results of the moderating effects.

	Model (3)					
EHC	0.0174 (0.59)	0.0181 (0.62)	−0.00467 (−0.15)			
HHC				0.244 *** (3.66)	0.260 *** (3.77)	0.222 *** (3.09)
ER1	0.0365 (1.42)			0.0287 (1.21)		
ER2		0.00397 * (1.82)			0.00472 ** (2.10)	
ER3			0.0161 (1.37)			0.0171 (1.52)
EHC * ER1	−0.00223 (−0.16)					
EHC * ER2		−0.000182 (−0.11)				
EHC * ER3			−0.0107 ** (−2.17)			
HHC * ER1				0.0374 (1.40)		
HHC * ER2					0.00106 (0.40)	
HHC * ER3						−0.0135 (−1.53)
_cons	2.847 * (1.91)	3.721 ** (2.41)	3.114 ** (2.10)	3.314 ** (2.32)	4.213 *** (2.85)	3.671 ** (2.55)
N	330	330	330	330	330	330
R2-within	0.9620	0.9621	0.9623	0.9639	0.9639	0.9638
R2-between	0.2799	0.2723	0.3121	0.2202	0.2057	0.2427
R2-overall	0.4755	0.4701	0.4996	0.4284	0.4179	0.4474
F-value	420.87	423.06	425.34	444.65	444.99	443.40
*p*-value	0.0000	0.0000	0.0000	0.0000	0.0000	0.0000

(Notes: t-statistics in parentheses. *** *p* < 0.01. ** *p* < 0.05. * *p* < 0.1. GTI—Green technology innovation; EHC—Educational human capital; HHC—Healthy human capital; ER1—Command-and-control environmental regulations; ER2—Market-incentivized environmental regulations; ER3—Public voluntary environmental regulations; PGDP—Regional economic development level; STR— Industrial structure; RD—R&D intensity; UR—Urbanization level).

**Table 5 ijerph-20-04803-t005:** Robustness test 1.

	Model (1)			Model (2)		
EHC	0.0273 ** (2.49)					
L3.EHC		0.0245 * (1.86)				
HHC			0.102 *** (4.90)			
L.ER1				0.00952 ** (2.07)		
ER2					0.00105 *** (3.07)	
ER3						−0.000877 (−0.51)
PGDP	0.0979 *** (3.20)	0.0247 (0.69)	0.0630 ** (2.08)	0.0752 ** (2.37)	0.0553 * (1.79)	0.0815 *** (2.67)
STR	0.00197 ** (2.45)	0.00419 *** (4.39)	0.00198 ** (2.44)	0.00221 *** (2.62)	0.00183 ** (2.18)	0.00205 ** (2.48)
RD	−0.0000146 ** (−2.00)	−0.0000286 ** (−2.57)	0.0000103 (1.43)	−0.0000221 *** (−2.89)	−0.0000256 *** (−3.33)	0.0000224 *** (−2.92)
UR	0.00399 ** (2.27)	0.00257 * (1.08)	0.0657 *** (2.45)	0.00367 ** (1.98)	0.00487 *** (2.60)	0.00420 ** (2.23)
_cons	0.00399 ** (2.27)	1.254 *** (4.51)	0.761 *** (3.22)	0.791 *** (3.26)	0.894 *** (3.78)	0.680 *** (2.95)

(Notes: z-statistics in parentheses. *** *p* < 0.01. ** *p* < 0.05. * *p* < 0.1. GTI—Green technology innovation; EHC—Educational human capital; HHC—Healthy human capital; ER1—Command-and-control environmental regulations; ER2—Market-incentivized environmental regulations; ER3—Public voluntary environmental regulations; PGDP—Regional economic development level; STR—Industrial structure; RD—R&D intensity; UR—Urbanization level).

**Table 6 ijerph-20-04803-t006:** Robustness test 2.

	Model (1)		Model (2)			
L3.EHC	0.0574 * (1.75)					
HHC		0.274 *** (3.75)				
L.ER1			0.0348 (1.32)			
ER2					0.00539 ** (2.34)	
ER3						0.00626 (0.53)
PGDP	0.0652 (0.25)	−0.215 (−1.15)	−0.0756 (−0.38)		−0.282 (−1.43)	−0.174 (−0.90)
STR	0.0181 *** (2.71)	0.0137 *** (2.87)	0.0154 *** (3.03)		0.0122 ** (2.51)	0.0133 *** (2.73)
RD	0.000119 ** (2.17)	0.000310 *** (6.49)	0.000200 *** (4.62)		0.000197 *** (4.72)	0.000218 *** (5.21)
UR	0.0511 *** (4.27)	0.0721 *** (8.17)	0.0670 *** (7.11)		0.0760 *** (8.36)	0.0721 *** (7.97)
_cons	1.508 (0.66)	2.652 * (1.70)	2.032 (1.21)		3.349 ** (2.04)	2.430 (1.51)
N	240	330	300		330	330
R2-within	0.9281	0.9556	0.9476		0.9542	0.9534
R2-between	0.2444	0.1936	0.2685		0.2514	0.2706
R2-overall	0.3528	0.3862	0.4178		0.4352	0.4510
F-value	212.95	408.50	330.56		396.24	388.83
*p*-value	0.0000	0.0000	0.0000		0.0000	0.0000
	Model (3)					
EHC	0.0310 (0.96)	0.0294 (0.94)	0.00974 (0.28)			
HHC				0.269 *** (3.69)	0.284 *** (3.77)	0.244 *** (3.10)
ER1	0.0361 (1.29)			0.0297 (1.14)		
ER2		0.00501 ** (2.11)			0.00585 ** (2.39)	
ER3			0.0165 (1.28)			0.0187 (1.53)
EHC * ER1	−0.00283 (−0.19)					
EHC * ER2		−0.000638 (−0.35)				
EHC * ER3			−0.0103 * (−1.92)			
HHC * ER1				0.0364 (1.25)		
HHC * ER2					0.000649 (0.22)	
HHC * ER3						−0.0152 (−1.58)
_cons	1.928 (1.20)	3.046 * (1.83)	1.910 (1.37)	2.632 * (1.64)	3.720 ** (2.30)	2.758 * (1.88)
N	330	330	330	330	330	330
R2-within	0.9539	0.9544	0.9542	0.9561	0.9565	0.9561
R2-between	0.2438	0.2354	0.2708	0.1913	0.1770	0.2118
R2-overall	0.4279	0.4217	0.4494	0.3833	0.3723	0.4021
F-value	344.20	348.52	347.12	362.76	366.24	362.85
*p*-value	0.0000	0.0000	0.0000	0.0000	0.0000	0.0000

(Notes: t-statistics in parentheses. *** *p* < 0.01. ** *p* < 0.05. * *p* < 0.1. GTI—Green technology innovation; EHC—Educational human capital; HHC—Healthy human capital; ER1—Command-and-control environmental regulations; ER2—Market-incentivized environmental regulations; ER3—Public voluntary environmental regulations; PGDP—Regional economic development level; STR—Industrial structure; RD—R&D intensity; UR—Urbanization level).

## Data Availability

Data will be made available on request.

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
