# Peer review of "The Impact of Human Capital on Green Technology Innovation—Moderating Role of Environmental Regulations"

_ijerph, 2023, doi:10.3390/ijerph20064803_

Round 1

Reviewer 1 Report

Thank you for the opportunity to review this interesting paper. I believe the topic is timely and relevant, and there is potential to add to the literature; however, some minor issues should be cleared up for the next step.

  1. Abstract
    • Please be more specific about the research purpose and justify the theoretical gap.
  2. Introduction
    • It seems that there is a lack of theoretical explanation. It is because the gap identification is not explained clearly and is limited in the current form. Also, there is very little evidence to support the authors' arguments on the gap justification. Could you please clarify the theoretical gaps that have not been discovered in the existing theories? What theory was employed in this study to fill the theoretical gap? Several studies use game theory as a core basis.
      • Wicki, S., & Hansen, E. G. (2019). Green technology innovation: Anatomy of exploration processes from a learning perspective. Business Strategy and the Environment28(6), 970-988.
  3. Literature review and hypothesis development
    • A theoretical review seems necessary. Please explain how the implications obtained through comparison, rather than listing related literature, will narrow the gap with existing theories. The explanations of the theoretical gaps are complexly listed. Therefore, the research question should be a little clearer.
    • The literature review should be conducted to the extent that hypotheses can be presented in a nuanced manner, but this manuscript is somewhat lacking in that respect.
    • H3 and H4 are understood as moderating effects. Please explain the situation and internal mechanism a little more. Each type of environmental regulation should be specified to increase/decrease the direct relationship between indep and dep. “how” is not clarified with logical reasonings. In this case, I suggest authors to focus on using examples or contextual evidences from previous articles.
    • Figure 1 is nice to see the whole picture of logics; but, as I commented, moderation effect logics are somewhat weak and missed.    
  4. Methodology
    • Add and make your research context richer.
    • Please document an appropriate extraction method for sample justification, generalization, and analytical technique for the integrated review.
    • It is necessary to show the relationship between the elements used in the study as a single framework or figure.
    • It is necessary to distinguish the estimator and verify it. Patent analysis is difficult with simple OLS. Negative binomial regression or zero-inflation should be used. And also, system GMM is needed to check the robustness due the previous year’s remained effect on existing patents.
    • Please put notes for abbreviations (e.g., EDU, HEL, ER, STR, etc.) under each table to make readers read it easily.
    • Fit indices should be added: adj-R2, R2_between, R2_overall, R2_within, log-likelihood, F-value and it’s p-value.
    • Are moderators mean-centered?
    • As for robustness, if you want to use alternative proxy, please add references. And, please discuss the meaning of what you figured out by using this test.
  5. Conclusion
    • The theoretical contribution of this study is somewhat weak. The theoretical contribution should be made clear of what has been extended in the existing literature. It is considered that the current theoretical implications are sufficient to make meaningful linkages of existing studies.
      • Please review the following papers and provide updates for theoretical and managerial contributions.
      • For consumer green innovation: Open for green innovation: From the perspective of green process and green consumer innovation, 11(12), 3234.
      • For green process/product innovation: How do intellectual property rights and government support drive a firm's green innovation? The mediating role of open innovation. Journal of Cleaner Production, 317, 128422.
      • For green managerial innovation: Structural relationships a firm's green strategies for environmental performance: The roles of green supply chain management and green marketing innovation, Journal of Cleaner Production, 356, 131877.

Author Response

Thank you very much for your time involved in reviewing our manuscript. Your constructive remarks and useful suggestions have significantly raised the quality of the manuscript and enabled us to improve the research. The revision is as follows.

Reviewer 2 Report

Review for ijerph

This is an interesting paper on the mechanisms driving the invention and diffusion of green technology in China. It fits well into the scope of the journal. At the same time, there are some issues the authors should address before publication.

The authors should strengthen ties to the existing literature on the diffusion of green technology. And they should add additional ties to existing work more generally.

This starts with the introduction. References strengthening the suggestion that China simultaneously has been promoting economic growth and environmental protection are lacking.

one is that environ-mental regulations can promote green technology innovation through the “innovation compensation” _effect, the other is that environmental regulations may inhibit green tech-nology innovation through the “compliance c_o_s_t_” _e_f_f_e_c_t_,_ _and the third is that different types of environmental regulations have different effects on green technology innovation.” P. 2

Please add references.

China has built the world's largest higher education system” P. 2

Whit respect to absolute numbers, maybe. With respect to enrollment rate? Is a relative measure not more relevant, when one tries to quantify social capital? Please be careful with these kinds of statements. On p. 6 the authors then introduce a relative measure, supporting my concerns with respect to the issue.

Theoretically, the authors heavily rely on the notion of human capital (P. 2). This makes sense from a perspective of innovation generation. Yet factors related to social capital rather than human capital drive the acceptance of behaviors, products, and technologies that are individually costly but collectively beneficial (Ostrom and Ahn 2009). This might be worth mentioning.

Ostrom, E., & Ahn, T. K. (2009). The meaning of social capital and its link to collective action. Handbook of social capital: The troika of sociology, political science and economics, 17-35.

H3, different regulations have different impacts, is rather blunt. Please be more specific here. What kinds of regulations likely promote, which likely constrain green technology diffusion?

A crucial point is the following question. Why do informal drivers of green technology diffusion seem to play no role in China? The authors find that formal but informal factors drive the innovation of green technologies in China. In other regions, informal factors, such as voluntary action, and informal norms sometimes – although not always – play a role. The authors should add a section addressing a potential explanation for to the discussion section. It seems that the distribution of preferences in the population (e.g. prosocial preferences, environmental awareness) (Nyborg 2020, Berger 2021, Berger et al. 2021), the adoption cost of new technology (Diekmann and Preisendörfer 2004), and features of the new technology (e.g. the signaling value) (Berger 2019, Griskevicius et al. 2010 all play a role. 

Berger, J. (2019). Signaling can increase consumers' willingness to pay for green products. Theoretical model and experimental evidence. Journal of consumer behaviour, 18(3), 233-246.

Berger, J. (2021). Social tipping interventions can promote the diffusion or decay of sustainable consumption norms in the field. Evidence from a quasi-experimental intervention study. Sustainability, 13(6), 3529.

Berger, J., Efferson, C., & Vogt, S. (2021). Tipping pro-environmental norm diffusion at scale: opportunities and limitations. Behavioural public policy, 1-26.

Diekmann, A., & Preisendörfer, P. (2003). Green and greenback: The behavioral effects of environmental attitudes in low-cost and high-cost situations. Rationality and Society, 15(4), 441-472.

Griskevicius, V., Tybur, J. M., & Van den Bergh, B. (2010). Going green to be seen: status, reputation, and conspicuous conservation. Journal of personality and social psychology, 98(3), 392.

Nyborg, K. (2020). No man is an island: social coordination and the environment. Environmental and Resource Economics, 76, 177-193.

Ostrom, E., & Ahn, T. K. (2009). The meaning of social capital and its link to collective action. Handbook of social capital: The troika of sociology, political science and economics, 17-35.

*Minor points

Add a blank space after (1), (2) and (3), same on p. 12, first paragraph.

Author Response

(The authors gave the same response as above.)

Round 2

Reviewer 1 Report

1. Please change EDU to ER1, 2, 3, or whatever the authors named it.

2. Please upload a clean version of the revised manuscript. 

Overall your endeavors to revise the manuscript deserves to be publishment. 

Author Response

Dear reviewer,

Many thanks again for your constructive remarks and useful suggestions. We have revised the content of the manuscript according to your comments as follows:

(1) Comment: Please change EDU to ER1, 2, 3, or whatever the authors named it.

Reply:

Thank you very much for your kind reminder. We have used EHC to represent educational human capital and HHC to represent healthy human capital. In addition, we have changed the EDU to ER1, ER2, ER3 in Figure 2.

(2) Comment: Please upload a clean version of the revised manuscript. 

Reply:

Yes! We have uploaded a clean version of the revised manuscript.

Best wishes,

Sincerely yours.